# The Molecular Mechanism of Epithelial–Mesenchymal Transition for Breast Carcinogenesis

**DOI:** 10.3390/biom9090476

**Published:** 2019-09-11

**Authors:** Chia-Jung Li, Pei-Yi Chu, Giou-Teng Yiang, Meng-Yu Wu

**Affiliations:** 1Department of Obstetrics and Gynecology, Kaohsiung Veterans General Hospital, Kaohsiung 813, Taiwan; 2Institute of Biomedical Sciences, National Sun Yat-Sen University, Kaohsiung 804, Taiwan; 3School of Medicine, College of Medicine, Fu Jen Catholic University, New Taipei 242, Taiwan; 4Department of Pathology, Show Chwan Memorial Hospital, Changhua 500, Taiwan; 5Department of Health Food, Chung Chou University of Science and Technology, Changhua 510, Taiwan; 6Department of Emergency Medicine, Taipei Tzu Chi Hospital, Buddhist Tzu Chi Medical Foundation, New Taipei 231, Taiwan; 7Department of Emergency Medicine, School of Medicine, Tzu Chi University, Hualien 970, Taiwan

**Keywords:** breast cancer, transforming growth factor-β, epithelial-to-mesenchymal transition, signaling pathway

## Abstract

The transforming growth factor-β (TGF-β) signaling pathway plays multiple regulatory roles in the tumorigenesis and development of cancer. TGF-β can inhibit the growth and proliferation of epithelial cells and induce apoptosis, thereby playing a role in inhibiting breast cancer. Therefore, the loss of response in epithelial cells that leads to the inhibition of cell proliferation due to TGF-β is a landmark event in tumorigenesis. As tumors progress, TGF-β can promote tumor cell invasion, metastasis, and drug resistance. At present, the above-mentioned role of TGF-β is related to the interaction of multiple signaling pathways in the cell, which can attenuate or abolish the inhibition of proliferation and apoptosis-promoting effects of TGF-β and enhance its promotion of tumor progression. This article focuses on the molecular mechanisms through which TGF-β interacts with multiple intracellular signaling pathways in tumor progression and the effects of these interactions on tumorigenesis.

## 1. Introduction

Breast cancer is a common cancer in women worldwide with increasing incidence and mortality rates. In Latin America, 200,000 women are diagnosed with breast cancer per year, with more than 52,000 deaths annually [1,2,3]. The high incidence and mortality rate of this disease have led to an increase in research meant to combat this public health issue. In order to accurately predict the clinical outcome of breast cancer, several types of scoring systems are used, based on histopathological appearance, anatomical location, molecular alterations, disease presentation, and clinical features. Moreover, in a recent study, molecular classification was revealed to be especially important in predicting clinical outcome, as it was associated with drug resistance [4]. According to current information, breast cancer can be divided into six major subgroups based on their molecular portrait including normal-like, HER-2 positive, luminal A and B type, basal-like, and claudin-low. The normal-like subgroup has an expression profile similar to that of noncancerous breast tissue. The overexpression of ErbB2, a receptor-like tyrosine kinase oncogene also known as human epidermal growth factor receptor 2 (HER-2), influences several signaling pathways and promotes dysregulated growth, oncogenesis, metastasis, and chemoresistance in breast cancer. The HER2 overexpression has been reported with poor prognosis, especially in patients without chemotherapy and target therapy [5]. The luminal A and B breast cancer subtypes generally express luminal cytokeratin 8/18 and the estrogen receptor, but at different levels. The luminal A subtype is generally characterized by higher estrogen receptor (ER) expression and lower HER-2 expression. In contrast with the luminal A subtype, luminal B breast cancer is usually characterized by lower ER expression and a higher Ki67 index, leading to advanced breast cancers with high proliferation rates and a worse prognosis. The basal-like subtype is characterized by the expression of biomarkers in the basal/myoepithelial cells of normal breast tissue such as cytokeratin 5/6, cytokeratin 14, cytokeratin 17, vimentin, P-cadherin, and p63 [6,7,8,9]. The claudin-low subtype is characterized by the low expression of cell–cell adhesion molecules including claudins 3, 4, and 7, occludin, and E-cadherin [10,11]. This subtype is also characterized by the presence of epithelial-to-mesenchymal transition (EMT) processes and stem cell-associated features [12]. The basal-like and claudin-low subtypes are commonly found in triple-negative breast cancer (TNBC), which is characterized by the lack of hormone receptors such as PR, ER, and HER-2, and is associated with higher recurrence and distant metastasis rates. The expression of estrogen receptor (ER) and the progesterone receptor (PR) are important predictive markers for hormone therapy [13]. These receptors can be used as targets for adjuvant endocrine therapy in order to regulate breast carcinogenesis. Patients who receive this type of therapy have been shown to have a better prognosis including overall survival, disease-free survival, and time to treatment failure [14]. On the other hand, the lack of PR expression in breast cancer leads to a more aggressive progression and a poorer prognosis. Due to the emergence of molecular analysis methods, the detailed mechanisms of tumorigenesis in undifferentiated phenotypes are essential in providing novel targets for treatment.

## 2. Epithelial-to-Mesenchymal Transition in Breast Cancer

Breast carcinogenesis is a complex, multiple step process, involving several mechanisms that mediate cell proliferation, differentiation, apoptosis, epithelial-to-mesenchymal transition, and angiogenesis [15]. In breast cancers with a poorly differentiated phenotype, the tumor cell is characterized by stem cell-like features, which arise due to the EMT process. This promotes the process of dedifferentiation and leads to a worse prognosis [16]. For example, EMT markers such as vimentin, N-cadherin, and cadherin-11, have been reported to be upregulated in triple-negative breast cancer (TNBC), thereby promoting extracellular matrix remodeling via matrix metallopeptidases (MMPs) and decreasing the expression of epithelial markers, finally leading to a poor clinical outcome [17]. In previous studies, invasion and metastasis were shown to be the major risk factors associated with a poor clinical outcome, which are also related to the EMT process [18,19,20,21]. The transcription factors that are involved in the EMT process such as SNAIL1/2, ZEB1/2, TWIST1/2, and FOXC1/2 play an important role in mediating embryogenesis and carcinogenesis by regulating the expression of E-cadherin (Table 1) [22]. Currently, the EMT program is divided into three types: embryogenesis, fibrosis, and tumorigenesis. Type 1 and 2 EMT contribute to organ development and tissue regeneration [23]. Type 3 EMT is involved in breast carcinogenesis and has been reported to be significantly associated with local invasion and distant metastasis [24,25]. It is also involved in regulating several cellular functions including cellular adhesion, migration, proliferation, differentiation, survival, and metastasis through several processes such as loss of epithelial polarity, detachment of the basement membrane, and acquisition of mesenchymal features [18,19,26]. Advanced breast cancer is often characterized as having stem cell-like features, which appear due to the EMT process. This includes loss of hormone receptors and cell–cell interaction proteins. In vitro, the estrogen knockdown reporter model of MCF-7 showed that the loss of ER expression was significantly associated with the EMT process, thereby promoting cell proliferation and migration by increasing the extracellular matrix and reducing matrix metalloproteases [27]. As such, EMT was thought to be an important step in carcinogenesis and the formation of distant metastasis [28,29]. In addition, the stem cell-like features induced by EMT were shown to contribute to drug resistance [30]. Several EMT-related signaling pathways play an important role in drug resistance in breast cancer cells. Cells undergoing EMT show similar cancer stem cell function including an increase in drug efflux pumps and anti-apoptotic effects. The two features increase drug resistance in cancer cells. Aggressive TNBC tumors such as metaplastic breast cancer are usually characterized by resistance to chemotherapy due to the activation of the EMT process, which is associated with worse outcomes [31]. The claudin-low subtype is also linked to metaplastic breast cancer due to the low expression of GATA3-regulated genes, which are involved in both the EMT process and cell adhesion. Notably, six critical components including TGF-β signaling, PI3K/AKT/mTOR signaling, regulatory factors, exosomes, and angiogenesis, were reported to regulate EMT by genetic or epigenetic alterations, thereby altering interaction with the extracellular matrix in breast carcinogenesis.

## 3. The Role of the TGF-β Pathway in Breast Cancer

Transforming growth factor-β (TGF-β), a multifunctional cytokine, directly regulates cell development, differentiation, homeostasis, proliferation, and transformation. TGF-β signaling plays an important role in the activation of the EMT and interacts with downstream signaling pathways in breast tumorigenesis [44]. The activation of TGF-β induces both canonical SMAD2/3-dependent signaling and non-SMAD signaling in order to promote the EMT process. In SMAD-mediated signaling, TGF-β directly binds to the membrane receptors, leading to the formation of the SMAD complex by activating SMAD2/3/4. In non-SMAD signaling pathways, TGF-β triggers the AKT/PI3K pathway, Ras/Raf/MEK/ERK signaling pathway, and Wnt/β-catenin signaling pathway in order to induce the expression of epithelial proteins [45,46]. Moreover, the TGF-β type I receptor interacts with the Src homology 2 domain-containing transforming protein 1 (SHC1) to activate the growth factor receptor-bound protein 2 (GRB2) and son of sevenless (SOS) in order to induce Ras/Raf/MEK/ERK signaling. In addition, TGF-β may phosphorylate Par6 directly via the type II receptor, thereby promoting the degeneration of RhoA via Smurf1 and inducing the dissolution of the tight junctions [47]. Par6 plays an important role in stress fiber formation, thereby regulating cell polarity and junction stability. In breast carcinogenesis, the partitioning defective 6 (PAR6) promotes the loss of polarity via TGF-β-dependent signaling and induces mesenchymal-like invasive mammary tumor cells. Notably, studies have shown that by blocking Par6 signaling, the EMT process can be curbed [48]. These results were confirmed by the formation of ZO-1-positive epithelium-like structures in breast carcinogenesis. Moreover, distant metastasis was also suppressed [20,48]. The TGF-β receptor also induced the expression of three Ras-related GTP-binding proteins, namely RhoA, RAC1, and CDC42, leading to cytoskeletal changes by regulating the actin cytoskeleton in response to extracellular signals [49]. In addition, TGF-β interacts with the PI3K/AKT pathway for translational regulation. In the study by Fei Huang et al. [50], HER2/EGFR signaling switched the TGF-β function in breast cancer to activate phosphorylation of Smad3 through AKT, promoting epithelial–mesenchymal transition and migration. TGF-β also interacts with Wnt signaling via β-catenin. In the study by Anders Sundqvist et al. [51], TGF-β triggered Wnt7a/7b via Smad2/3, enhancing TGF-β-induced EMT of the mammary epithelial cells, and the components of the WNT signaling pathway were enriched within the late TGF-β target genes. Moreover, glycogen synthase kinase-3β (GSK3β) inhibits the β-catenin in the nucleus and activates the lymphoid enhancer-binding factor 1 (LEF) and T cell factor (TCF), thereby inducing the EMT process [52]. In an inducible c-fos estrogen receptor (FosER) cell model, β-catenin and TGF-β signaling cooperated to induce a mesenchymal phenotype during the EMT process. Inhibition of both signaling pathways in FosER cells led to a reversion from a mesenchymal phenotype to a polarized epithelial phenotype [53].

## 4. Relevant Regulatory Factors of EMT

The activation of EMT is characterized by the decrease in the expression of epithelial cell markers such as E-cadherin, occludin, tight junction proteins, and desmoglein, which are inhibited by transcription, resulting in a loss of adhesion between cells. Moreover, it is accompanied by a simultaneous upregulation of mesenchymal cell markers including N-cadherin, cytokeratin, actin, and fibronectin [54]. E-cadherin is an important molecule in the development of EMT and its downregulation is closely related to tumor invasion and metastasis. Studies have found that disruption of E-cadherin expression in intercellular adhesions causes tumor cell metastasis [55]. In addition, E-cadherin loss also induced multiple transcription factors such as Twist for E-cadherin loss-induced metastasis, effectively inducing EMT and causing tumor cell metastasis. The transcription factors including Snail, Slug, ZEBl/ZEB2, and Twistl/Twist2 directly act on the E-box sequence of the CDH1 promoter to inhibit transcription and promote EMT activation [56]. Other transcription factors such as SIX Homeobox 1 (Six1), Goosecoid Homeobox (GSC), and Forkhead Box C2 (FOXC2) also promote EMT by indirectly inhibiting CDH1 [57,58]. However, the detailed mechanism is unclear. In the MCF7 breast cancer cell line, Six1 induces tumor-associated EMT by activating the TGF-β signaling pathway in order to downregulate E-cadherin. Furthermore, the SOX4 gene was found to be abnormally overexpressed in clinical specimens of human breast cancer. After overexpressing SOX4, human mammary epithelial cells leads to mesenchymal cell characteristics, enhanced cell migration as well as invasion [59]. SOX4 promotes EMT activation by upregulating the expression of the epigener of zeste 2 polycomb repressive complex 2 subunit (EZH2) [60]. Moreover, the downregulation of cellular communication network factor 6 (CCN6) is associated with EMT activation in breast cancer cells as well as with axillary lymph node metastasis in breast cancer. Knockdown of CCN6 in breast cancer cells upregulates Snail and ZEB1 expression at the RNA and protein level by activating the IGF1 receptor signaling pathway. It also indirectly inhibits the expression of epithelial cell markers such as E-cadherin and entry to initiate EMT activation [61]. In addition, inhibition of the growth family member 5 (ING5) leads to a decrease in the EMT conversion of breast cancer cells by inhibiting the P13K/AKT signaling pathway [62]. Moreover, studies have shown that cytoplasmic polyadenylation element binding protein 1 (CPEB1) inhibits breast cancer metastasis by reducing the expression of matrix metallopeptidase 9 (MMP9) mRNA [63].

EMT activation is regulated by precise intracellular signal transduction mechanisms, where a variety of extracellular signals bind to specific receptors on the cell surface. EMT is accomplished by transducing the signal into the cell and activating the relevant transcription factors in order to regulate gene expression through the intracellular signal transduction pathway. The occurrence of EMT involves multiple signaling pathways including TGF-β, PI3K/AKT, Ras/MAPK, Wnt, Notch, and Hedgehog pathways. These signaling pathways function by activating related transcription factors. For example, TGF-β activates Snail/2, Twist1, and ZEB1/2, and upregulates FOXC2, thereby inducing EMT activation [54]. In addition, the Notch, Hedgehog, and Wnt pathways act on EMT by activating Snail1/2. TGF-β, a canonical and non-canonical Wnt3 signaling pathway, synergistically induces EMT in cells, followed by the autocrine maintenance of interstitial cell status [64]. Studies have shown that downregulation of endogenous autocrine signaling inhibitors in epithelial cells induces cell-activated EMT processes including inhibitors of the TGF-β Receptor Type I (TGFBRI) (A83-01, SB435142) focused on TGF-β as well as canonical and non-canonical Wnt signaling. Conversely, the activation of specific signaling pathway inhibitors disrupts autocrine signaling in cells (Figure 1) [65]. In the study by Christina Scheel et al. [65], two TGFBRI inhibitors (A83-01, SB435142) disrupts the autocrine signaling result, which showed that due to these changes, the ability of primary mammary epithelial cells to migrate and self-renew was inhibited, and tumor formation and metastasis induced by the transformed derivatives were reduced. Recent studies in human breast cancer patients have demonstrated that TGF-β mediates the conversion of epithelial cells to stromal cells and that TGF-β originates from platelets [66,67].

## 5. Exosomes in EMT

In breast cancer and various other cancer studies, many regulatory factors that induce the activation of type III EMT also play an important role in type I EMT. This suggests that inappropriate activation-related regulatory factors in healthy individuals (e.g., transcription factors Twistl, Sixl, Snail, LBXl, and signaling pathways Wnt, TGF-β, etc.) will lead to the activation of type III EMT, thereby resulting in the occurrence and development of malignant tumors [68]. In the EMT field, exosomes are currently being intensely researched to determine their role in breast cancer invasion. In order for metastasis to occur, cells must communicate with their local environment to initiate growth and invasion. By transferring molecules such as mRNA, miRNA, and proteins between cells, exosomes have become important mediators of cell-to-cell signaling. Many studies have demonstrated that breast cancer cells promote tumor invasion and metastasis by transmitting molecules or signals through exosomes [69,70]. Moreover, breast cancer cells with metastatic potential secrete exosomes of chemokines with different protein characteristics which stimulate cell movement, indicating that the released exosomes can play a positive role in metastasis [71]. The exosomes further stimulate cancer cell invasion by direct feedback from extracellular matrix metalloproteinase inducers that rely on being highly glycosylated [72,73]. By using a microfluidic chip to quantify exosomes, circulating EpCAM-positive exosomes were detected in six breast cancer patients and three healthy controls and were found to be comparative with healthy controls. The level of EpCAM-positive exosomes in these breast cancer patients was significantly increased [74]. EpCAM also supports the regulation of EMT by inhibiting ERK activity and expressing SNAIL2, which defines a double negative feedback loop [75]. Exosome composition is significantly different between untransformed and transformed cells [76]. In addition, there is increasing evidence that tumor-derived exosomes (TDEs) and tumors from microenvironment (TME) are significantly associated with regulating tumor growth and survival as well as tumor invasion, angiogenesis, and metastasis.

## 6. Angiogenesis in EMT

Breast cancer invasion and metastasis are closely aligned with angiogenesis. Vascular endothelial growth factor (VEGF) is the most potent angiogenic factor currently discovered, which can specifically stimulate endothelial cell proliferation, promote vascular permeability, and provide a matrix for vascular endothelial cell migration and tumor cell metastasis [77]. TGF-β1 is a potent inducer of vascular endothelial growth factor (VEGF) in tumor cells. It is also involved in the tumor microenvironment, regulating tumor cell invasion and angiogenesis. Using bioluminescence imaging technology, studies have found that inducible VEGF may promote cell proliferation and reduce apoptosis induced by oxidative stress through autocrine mechanisms. Moreover, immunohistochemistry analysis confirmed that the induction of VEGF overexpression promoted cell survival and tumor neovascularization [78]. Notably, the expression levels of VEGF-A and VEGF-C in cancer cells of patients with breast cancer lymph node metastasis were significantly higher than those in non-metastatic breast cancer. The key role of VEGF-A is to promote breast cancer cell proliferation and accelerate tumor growth, while VEGF-C mainly promotes the formation of lymphatic vessels in the peri-cancer region, which is important for the invasion and metastasis of breast cancer [79,80].

## 7. ECM Remodeling during EMT

Breast cancer invasion is associated with the expression of matrix metalloproteinases (MMPs), which degrade the extracellular matrix and basement membrane, allowing tumor cells to penetrate and pass through. This is thought to be closely related to local invasion and distant metastasis of breast cancer cells. Moreover, a higher expression of MMPs was associated with a poorer prognosis in breast cancer patients, with MMP-2 and MMP-9 being the major enzymes involved in this process [81,82]. Previous studies confirmed that the gene-transfected MMPs breast cancer cell line MDA-MB-436 was inoculated into nude mice and induced the over production of MMP-2 and MMP-9. In mice containing cells transfected with MMP, the metastatic ability of cancer cells was significantly enhanced, indicating that MMP-2 and MMP-9 may promote invasion and metastasis of breast cancer [82]. In addition, the MMP-2 gene was transfected into another breast cancer cell line, MDA-MB-231. The invasive ability of the cell line was also found to increase. When the cell line was implanted in the mammary gland of nude mice, the tumor cells rapidly proliferated, while the metastasis rate of tumors in various parts of the viscera significantly increased. This suggests that MMP-2 not only accelerates the proliferation of cancer cells, but is also closely related to tumor invasion and metastasis [83]. Unlike normal human breast tissue which has little to no expression of MMP-9, triple negative HER-2 positive and lymph node metastatic breast cancer cells often overexpress MMP-9 [84]. The overexpression of MMP-9 is associated with higher rates of metastasis, shorter recurrence latency, and shorter post-recurrence survival as well as with breast cancer invasiveness [85,86,87]. The ECM is characterized by a gradual change during cancer progression.

## 8. PI3K/AKT/mTOR Signaling in EMT

PI3K CA (phosphatidylinositol 3-kinase catalytic subunit alpha) is a 34 kb gene located on chromosome 3q26.3. Mutation of PIK3CA triggers the activation of proto-oncogenes [88,89]. There are three mutation sites, namely E542K, E545K, and H1047R, which are mainly involved in the activation of PI3K mutations. Once the gene is mutated, the cells will phosphorylate AKT, P70S6K, and 4EBP1, respectively. P70S6K and 4EBP1 are regulated by mTOR and affect protein synthesis, while AKT activates and initiates the growth and transformation of mammary epithelial cells and inhibits apoptosis [90]. Although the three mutation sites affect the PI3K pathway differently, all three mutation sites lead to an increase in PI3K activity and initiate the signal transduction of carcinogenic mutant genes. *PTEN* is located on chromosome 10q23 and acts as a tumor suppressor gene through the action of its phosphatase protein product. It dephosphorylates PI (3,4,5) P3, a product of PI3K, and converts PIP3 into PIP2. Deletion of the tumor suppressor gene *PTEN* triggers overactivation of the PI3K pathway, which in turn leads to accumulation of intracellular PIP3 and activation of downstream factors (PDK1 and Akt/PKB). In addition, the sensitivity of the PTEN mutation to apoptotic factors is reduced. Studies have shown that *Pten* knockout mouse models form basal-like breast cancer, and that *Pten* heterozygous loss is associated with human basal-like breast cancer formation. However, the detailed mechanisms underlying these processes need to be further clarified [91]. Notably, in primary breast cancer and metastatic breast cancer, the overall level of *PIK3CA* mutation is approximately 40.4%, while the overall level of PTEN loss is approximately 30.4% [92]. Previous studies have also shown that PI3Kβ-silenced breast cancer cells reduce metastasis by regulating extravasation, which may be associated with disrupting macrophage-induced tumor cell invasion and reducing the metastasis of breast cancer cells [93].

The PI3K signaling pathway involves multiple genes. When the proto-oncogene activation or tumor suppressor gene are inactivated, molecular signaling is over-activated, thereby regulating the downstream signaling pathway and triggering tumor formation through multiple mechanisms: (1) Promoting excessive cell proliferation: The phosphorylation of 4EBP1 after mTOR activation separates eIF4E from 4EBP1 and binds to other translation initiation factors to initiate translation of proliferation-related proteins, thereby accelerating cell proliferation; and (2) Inhibit apoptosis: AKT inhibits apoptosis-related proteins through phosphorylation such as the proapoptotic factor BAD, proteolytic enzyme caspase 9, and the Forkhead family transcription factor FKHR, which inactivates the Fas ligand, thereby impeding the normal transmission of apoptotic signaling pathways [94]. In addition, AKT activates phosphorylation of the tumor suppressor gene TSC2, which abolishes the inhibition of Rheb by TSCl/2, and then activates mTOR, which is involved in protein synthesis, to exert an anti-apoptotic effect [95].

## 9. Tumor Microenvironment and EMT Formation

The tumor microenvironment contains immune system cells, tumor vasculature, and lymphatic cells as well as fibroblasts, pericytes, and sometimes adipocytes. The tumor microenvironment also includes a matrix that interacts with the area surrounding cells. [96]. As the tumor progresses, the tumor microenvironment effectively blocks the infiltration of cytotoxic leukocytes from the host, promotes the inflammatory response, and recruits tumor-infiltrating lymphocytes to adapt to the host response [96]. Extracellular matrix (ECM) constitutes a scaffold of tissues and organs that provides biochemical and basic structural support for its cellular components. TGFβ secreted by tumor cells, related fibroblasts, or immune cells can induce new ECM synthesis, based on ECM remodeling by metalloproteinases and promote phenotypic changes in cell invasion [22]. TGFβ can induce the expression and stability of several ECM components. The TGFβ pathway regulates various ECM genes through SMAD and MAP-kinase signaling. Induction of various ECM remodeling enzymes in breast cancer promotes EMT signaling pathways and metastasis. The different pathways regulated by ECM during remodeling in breast cancer development are Wnt, PI3K, AKT, ERK, JNK, Src-FAK, etc. [97,98,99].

Recent studies have found that the inflammatory factors TNF-α and IL-1β in the tumor microenvironment stimulate normal breast epithelial cells adjacent to the cancer to cause structural remodeling and EMT activation, leading to malignant transformation of normal tissues and recurrence of disease [96,100,101]. In the tumor environment, the increase of inflammatory cytokines (TNF-α, IL-6, and LPS) and ROS expression under oxidative stress is crucial for the induction of the NF-κB pathway, and NF-κB can also directly activate the expression of potent EMT inducers including Snail and ZEB factors [102]. NF-κB has been found to inhibit the expression of the epithelial-specific gene E-cadherin and induce the expression of mesenchymal-specific gene vimentin. Snail is one of the important transcription factors for epithelial phenotype loss to inhibit E-cadherin expression, and NF-κB has been found to induce Snail expression, resulting in the downregulation of E-cadherin [103]. NF-κB also upregulates the transcription factors ZEB1 and ZEB2, leading to the inhibition of E-cadherin expression during EMT [104]. While the cells have entered the EMT process, blocking NF-κB activity leads to partial reversal of the mesenchymal phenotype [101].

## 10. Conclusions

During the EMT, tumor cells acquire invasive traits through overexpression, oncogene mutations, and inhibition of tumor suppressors, leading to aberrant expression of signaling pathways, further leading to tumor cells undergoing distant metastasis or migration to other organs after EMT. However, EMT is not an irreversible process, and reversing or inhibiting EMT may be an effective way to inhibit tumor cell migration or distant metastasis. Therefore, further study of EMT-related regulatory factors not only helps to better understand the mechanism of the EMT process in tumor cells, but also provides a new perspective for understanding the mechanism of tumor metastasis and recurrence and provides new clues for the treatment of tumors.

## Figures and Tables

**Figure 1 biomolecules-09-00476-f001:**
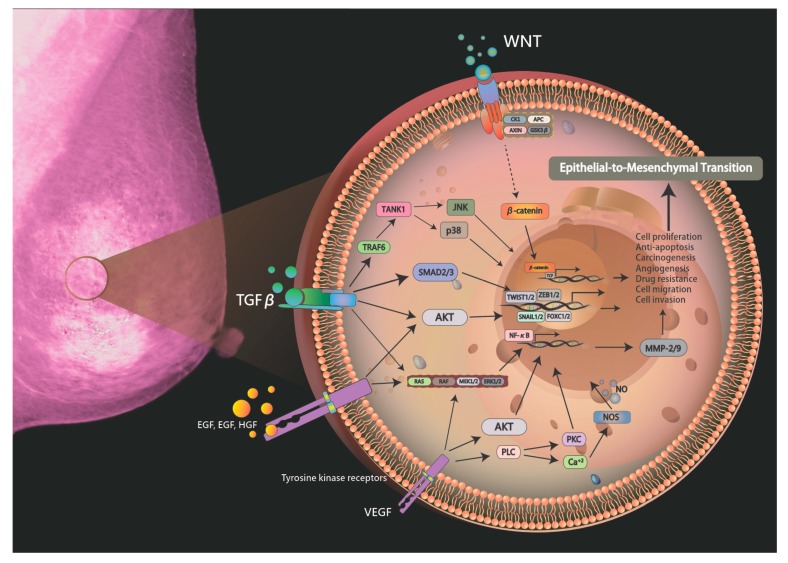
Schematic drawing presents the detailed signaling pathways of breast carcinogenesis. The exposure to hepatotoxic agents triggers gene mutation via signaling pathways including Raf-MEK-ERK pathway, PI3-kinase/Akt, protein kinase C/NF-κB, Src, and Wnt/β-catenin pathways.

**Table 1 biomolecules-09-00476-t001:** List of epithelial-to-mesenchymal transition regulators in cancer progression.

Family	Transcription Factor	Role	Ref.
Zinc-finger domain	SNAIL	Snail blocks the cell cycle and confers resistance to cell death.	[32]
	SLUG	Downregulation of E-cadherin expression occurs during the EMT, a process also exploited by invasive cancer cells.	[33]
	ZEB1	Represses E-cadherin promoter and induces EMT by recruiting SMARCA4/BRG1.	[34]
	ZEB2	ZEB2 protein is involved in chemical signaling pathways that regulate early growth and development.	[35]
bHLH	TWIST1	Overexpression of TWIST1 induces EMT, a key process in the metastasis formation of cancer.	[36]
FOX	FOXC1	FOXC1 partially promotes tumor metastasis by regulating EMT programs to support microvascular invasion, thereby increasing angiogenesis.	[37]
	FOXC2	Transcriptional activator that are upregulated in breast cancer.	[38]
Homeobox	SIX1	Six1 can promote the metastasis of human tumors, and the increased expression of Six1 can be used as an indicator for predicting breast cancer metastasis.	[39]
	LBX1	LBX1 is upregulated in the unfavorable estrogen receptor (ER)/progesterone (PR)/HER2 triple-negative basal-like subtype.	[40]
cadherin	E-cadherin	E-cadherin an active suppressor of invasion and growth of many epithelial cancers.	[41,42]
	N-cadherin	It is dependent on its association with the actin-cytoskeleton and is mediated through interactions with catenin proteins.	[43]

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
