# Peer review of "The Molecular Mechanism of Epithelial–Mesenchymal Transition for Breast Carcinogenesis"

_biomolecules, 2019, doi:10.3390/biom9090476_

Round 1
Reviewer 1 Report
Brief Summary: Liet alprovided a review of the role of the TGFb involved in EMT process in breast cancer carcinogenesis. Authors addresses the possible factors and multiple TGFb-related intracellular signaling pathways in breast cancer, which may result in tumor progression and high potential to obtain the resistance against therapies. Authors highlights a variety factors involved in regulation of EMT process and concludes the relation to metastasis process and clinical outcome. By organizing and clarifying these factors and the regulation pathways associated with the TGFb, might improve current therapy.
To highlight the importance of EMT process and TGFb-related signal pathways in tumor progression, authors summarized current study of factors involved in activation of EMT process and role of TGFb in pathways. Besides, potential communication role of exosome in EMT is also discussed. Microenvironment factors involved in degrading extracellular matrix (ECM) to promote metastasis is also elaborated. Current studies about the role of angiogenesis and PI3K/AKT/mTOR signaling pathways in EMT are discussed. Finally, factors in tumor microenvironment in stimulation of EMT process is also mentioned. EMT process is resulted from multiple pathways and factors, which then results in tumor migration and metastasis. Since EMT is not an irreversible process, the more we know about the mechanisms, the more potential applications in anticancer therapy will be revealed.
Broad Comments:
Overall, this paper includes comprehensive description. One suggestion could be considered in this review:
1. This paper is heavily focused on the discussion of EMT processes, which is a process related to multiple signal pathways, including TGFb. However, the title seems to address more focus on the TGFb-related mechanisms in breast carcinogenesis, which is a part of EMT process. The title needs to be more focus on the overall elaboration throughout the paper.
Specific Comments:
1. This is a manuscript of review. So the overall organization and visualization are key points for readers to make it easy to understand and follow. It would be helpful if authors can make tables to connect EMT to the related pathways and phenotype in cancer.
2. The Figure 1 is the overall signaling pathways of endometrial carcinogenesis; however, it is not readable. Simplified the pathways in the cell would be easier to read.
Author Response
Comments and Suggestions for Authors
Reviewer 1:
Brief Summary: Liet alprovided a review of the role of the TGFb involved in EMT process in breast cancer carcinogenesis. Authors addresses the possible factors and multiple TGFb-related intracellular signaling pathways in breast cancer, which may result in tumor progression and high potential to obtain the resistance against therapies. Authors highlights a variety factors involved in regulation of EMT process and concludes the relation to metastasis process and clinical outcome. By organizing and clarifying these factors and the regulation pathways associated with the TGFb, might improve current therapy.
To highlight the importance of EMT process and TGFb-related signal pathways in tumor progression, authors summarized current study of factors involved in activation of EMT process and role of TGFb in pathways. Besides, potential communication role of exosome in EMT is also discussed. Microenvironment factors involved in degrading extracellular matrix (ECM) to promote metastasis is also elaborated. Current studies about the role of angiogenesis and PI3K/AKT/mTOR signaling pathways in EMT are discussed. Finally, factors in tumor microenvironment in stimulation of EMT process is also mentioned. EMT process is resulted from multiple pathways and factors, which then results in tumor migration and metastasis. Since EMT is not an irreversible process, the more we know about the mechanisms, the more potential applications in anticancer therapy will be revealed.
Broad Comments:
Overall, this paper includes comprehensive description. One suggestion could be considered in this review:
This paper is heavily focused on the discussion of EMT processes, which is a process related to multiple signal pathways, including TGFb. However, the title seems to address more focus on the TGFb-related mechanisms in breast carcinogenesis, which is a part of EMT process. The title needs to be more focus on the overall elaboration throughout the paper.Response: We thank the reviewer for checking our article in detail and indicating the missing information. As the Reviewer’s comment, we have revised the title.
The Molecular Mechanism of Epithelial–Mesenchymal Transition for Breast Carcinogenesis
Specific Comments:
This is a manuscript of review. So the overall organization and visualization are key points for readers to make it easy to understand and follow. It would be helpful if authors can make tables to connect EMT to the related pathways and phenotype in cancer.Response: We thank the reviewer for checking our article in detail and indicating the missing information. As the Reviewer’s comment, we have made a table to connect EMT to the related pathways and phenotype in breast cancer.
The Figure 1 is the overall signaling pathways of endometrial carcinogenesis; however, it is not readable. Simplified the pathways in the cell would be easier to read.Response: We thank the reviewer for checking our article in detail and indicating the missing information. As the Reviewer’s comment, we have simplified the pathways in our figure 1
Reviewer 2 Report
The topic of this review article is quite important to the field of tumor metastasis and the authors have done a good job of highlighting signaling pathways and transcription factors previously shown to be involved in the EMT process. Although the general ideas are present in this paper, this manuscript does need quite a bit of work to have a larger impact. I am attaching my comments and suggestions for improving the manuscript. The authors need to look at the structure of each section and make sure there is an intro, body, and conclusion. Many of the sections do not have concluding summaries and end abruptly. Additionally the authors are encouraged to be more specific in their descriptions (upregulation/downregulation instead of alteration/modulation) and reference more data from their primary references.

Author Response
We thank the reviewer for checking our article in detail and indicating the missing information. As the Reviewer’s comment, we have revised our article based on the Reviewer’s comments and suggestions. Please, recheck our article and attachment. Thanks.

Round 2
Reviewer 2 Report
I am enthusiastic about the topic of the paper, but there are significant changes that still need to be made, particularly in the style of the paragraphs. Some paragraphs are lengthy with unrelated topics in the same paragraph. This second draft has improved from the first version and I have made my suggested changes in the attached PDF.

Round 3
Reviewer 2 Report
One small sentence change in attached document.

Author Response
We thank the reviewer for checking our article in detail. As the Reviewer’s comment, we have revised the manuscript.
Please see the attached file.
Thanks
